

# Characterization of phenylalanine ammonia-lyase genes facilitating flavonoid biosynthesis from two species of medicinal plant *Anoectochilus*

Lin Yang[1], Wan-Chen Li[2], Feng-ling Fu[2], Jingtao Qu[2], Fuai Sun[2], Haoqiang Yu[2] and Juncheng Zhang[1]

[1] Sanming University, Medical Plant Exploitation and Utilization Engineering Research Center Fujian Province University, Sanming, China

[2] Sichuan Agricultural University, Maize Research Institute, Chengdu, China

## ABSTRACT

**Background**. *Anoectochilus roxburghii* and *Anoectochilus formosanus*, belong to the *Anoectochilus* genus, have been used for Chinese herbal drugs as well as health food. Phenylalanine ammonia-lyase (PAL), the key enzyme in primary metabolism and phenylpropanoid metabolism, produces secondary metabolites (flavonoids) in plants, which are beneficial for the biosynthesis of phenylpropanoid metabolites.

**Methods**. The *PAL* genes were cloned from *A. formosanus* and *A. roxburghii* according to our previous transcriptomic analysis. The *PALs* were introduced into pCAMBIA2300-35S-PAL-eGFP to generate 35S-PAL-eGFP. The constructs were further used for subcellular localization and transgenic *Arabidopsis*. The expression of *AfPAL* and *ArPAL* under precursor substance (L-Phe), NaCl, UV, and red-light were analyzed by real-time quantitative PCR (RT-qPCR).

**Results**. *AfPAL* and *ArPAL*, encoding 2,148 base pairs, were cloned from *A. formosanus* and *A. roxburghii*. The subcellular localization showed that the ArPAL and AfPAL were both localized in the nucleus with GPF. Quantitative RT-PCR analysis indicated that the *ArPAL* and *AfPAL* genes function in the phenylalanine pathway as well as response to induced conditions. Overexpression of the *AfPAL* and *ArPAL* could increase flavonoids and anthocyanin content in the transgenic *Arabidopsis*.

**Discussion**. The results suggest that *AfPAL* and *ArPAL* play a crucial role in the flavonoid biosynthesis in *Anoectochilus*. Also, our study provides new insights into the enrichment of secondary metabolites of traditional Chinese medicines *A. formosanus* and *A. roxburghii*, which can improve their medicinal active ingredients and be used for drug discovery in plants.

## INTRODUCTION

*Anoectochilus* is a specie of orchidaceae family and possesses various pharmaceutical constituents, which plays an important role in cancer treatment (*Lang et al., 1999*; *Shyur et al., 2004*; *Yang et al., 2014*; *Yu et al., 2017*). Currently, *Anoectochilus formosanus* and

Corresponding authors
Haoqiang Yu, yhq1801@sicau.edu.cn
Juncheng Zhang, 19900204@fjsmu.edu.cn

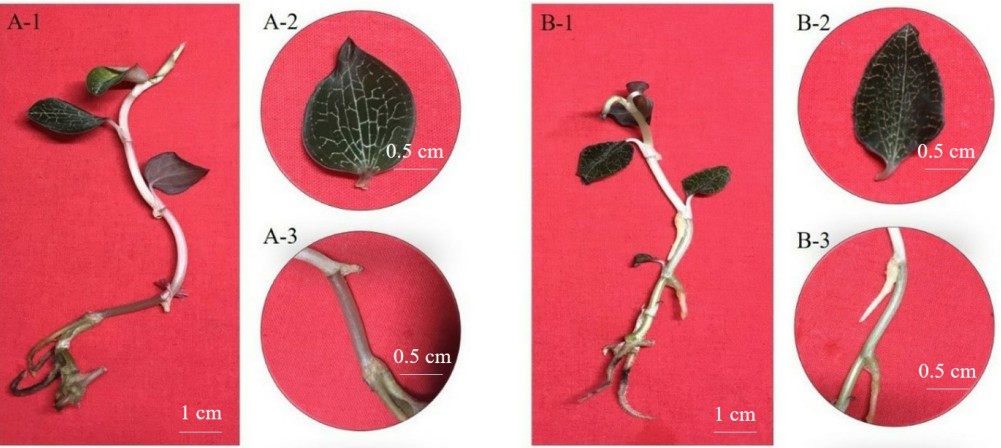

**Figure 1 The morphological characteristics of *A. formosanus* and *A. roxburghii*.** (A) and (B) represent *A. formosanus* and *A. roxburghii*; 1, 2 and 3 represents complete stool, leaf and stem; The leaf and stems of *A. formosanus* were white and pink, while those of *A. roxburghii* were golden yellow and green.

*Anoectochilus roxburghii* which are obviously different in morphology, are both widely used in cultivation or tissue culture for rapid propagation (Fig. 1, *Du et al., 2000*; *Shiau et al., 2002*; *Zhang et al., 2015*). However, the synthesis and catabolism of the pharmaceutical constituents including flavonoid, polysaccharides, glycoside derivative kinsenoside, and steroids in the cultivated *Anoectochilus* is primary and produced by secondary metabolites (*Dai et al., 2009*), which is different from wild plants (*Du et al., 2000*). It's significant to promote the accumulation of flavonoids of two medicinal plants *Anoectochilus* in artificial cultivation or tissue culture.

Phenylalanine ammonia-lyase (PAL, EC 4.3.1.24) is essential for connecting of primary and phenylpropanoid metabolism in plants. The PAL controls the speed of the first step in the biosynthesis of phenylpropanoid metabolites, the nonoxidative deamination of phenylalanine to trans-cinnamic acid and ammonia (*Lois et al., 1989*; *Nugroho, Verberne & Verpoorte, 2002*). Subsequently, phenylpropanoids will produce several secondary metabolites, such as flavonoid, phytohormone, anthocyanin, lignin, phytoalexin, and benzoic acid (Fig. S1, *Jorrin & Dixon, 1990*; *Jin et al., 2013*). To date, the *PAL* genes have been cloned and characterized by homologous amplification and rapid amplification of cDNA ends (RACE) from variant medicinal plants, such as *Dendrobium* (*Jin et al., 2013*), *Artemisia annua* (*Zhang et al., 2016a*; *Zhang et al., 2016b*), *Fagopyrum tataricum* (*Li et al., 2012*) and *Ginkgo biloba* (*Cheng et al., 2005*). However, genome information is not available for homologous amplification for any of the *Anoectochilus* species. The expression of the *PAL* gene and activity of the PAL protein are found to be responsive to light quality, salinity, drought, wounding, and related to secondary metabolites accumulation in other plants (*Nakazawa et al., 2001*; *Zhang et al., 2012*; *Zhang et al., 2016a*; *Zhang et al., 2016b*).

Given the importance of *A. formosanus*, *A. roxburghii*, and the active compounds in these *Anoectochilus*, it is essential for functional studies into the medicinal plant. In the present study, the *PAL* genes were cloned from *A. formosanus* and *A. roxburghii* according

to our transcriptional analysis. After bioinformatics analysis, the expression of the *AfPAL* and *ArPAL* genes in response to precursor substance (L-Phe), NaCl, UV, and red-light were detected by real-time quantitative PCR (RT-qPCR), respectively. The subcellular localization and heterologous expression of the *AfPAL* and *ArPAL* genes were performed. These results demonstrate that the *AfPAL* and *ArPAL* genes play an important role in flavonoids biosynthesis in *Anoectochilus*.

## MATERIALS & METHODS

### Sample preparation

The seedlings of *A. formosanus* and *A. roxburghii* were surface sterilized using 10% NaClO for 5 mins and plated onto MS medium in a chamber under a 12 h light/12 h dark at 28 °C and 60–80% humidity condition. The 4-month-old seedlings were transferred into a plastic mesh grid for aquaculture with Hoagland's nutrient solution. The seedlings were transferred into a plastic mesh grid for aquaculture. Phe and NaCl were added into the nutrient solution with a final concentration of 4 mg/L and 100 mmol/L. The seedlings were also transplanted into plastic pots cultivated with nutritional soil and vermiculite (3:1), and then induced under a 253.7 nm UV and 650 nm red light. The leaves samples were collected from each replicate at 0 (control), 0.5, 1, 2, 4, 8, 12 h of the Phe, NaCl, UV, and red-light induction, respectively. The RNA was extracted with Qiagen RNeasy plant mini kit (Qiagen, China), following cDNA were revere transcribed by PrimeScript RT reagent Kit (Takara China).

### Cloning of the *PAL* gene

The open reading frame (ORF) of the *AfPAL* and *ArPAL* gene from cDNA were amplified based on the annotation of RNA-seq by specific primers (5′-ATGGACCATGCTAGGGAG AACG-3′/5′-CTAGCAAATAGGGAGAGGAGCTTCA-3′) (http://www.premierbiosoft. com/primerdesign/). The amplified fragments were purified using Universal DNA Purification Kit (Tiangen, China), added dATP in the tail of sequences using the TaKaRa TaqTM (TaKaRa, China), cloned into pMD19-T vector (TaKaRa, China), and sequenced by Shanghai Sangon Biotech Co., Ltd (China).

### Bioinformatic analysis

The sequencing of the *AfPAL* and *ArPAL* genes were aligned for gene structure using blast on NCBI website (http://www.ncbi.nlm.nih.gov) and used for the analysis of physical and chemical properties, secondary structure, functional domains, and genetic structure of the putative proteins by using ProtParam (http://web.expasy.org/protparam), GOR IV (http://npsa-pbil.ibcp.fr/cgi-bin/npsa_automat.pl?page=npsa_gor4.html), TMHMM Server v. 2.0 (http://www.cbs.dtu.dk/services/TMHMM-2.0/) and SWISS-MODEL (https://swissmodel.expasy.org/) software or databases, respectively. Phylogenetic analysis among the putative amino acid sequences of the AfPAL and ArPAL proteins at the NCBI database were analyzed using the method of maximum likelihood of 1,000 bootstrap replicates by using MEGA7.0 software (https://www.megasoftware.net/). The evolutionary distances were computed by using the Poisson correction method.

## Vector construction

A pair of homologous arms (the lowercase bases) primers (5′-catttggagaggacagggtacccggg ATGGACCATGCTAGGGAGAACG-3′/5′-tcgcccttgctcaccatggtactagtGCAAA TAGGGAGAGGAGCTTCA-3′) was designed to amplify the ORFs of the *AfPAL* and *ArPAL* genes without termination codon for homologous recombination. The amplified PCR were inserted into the expression vector pCAMBIA2300 using CloneExpress One Step Cloning Kit (Vazyme, China) respectively, to generate a set of expression vectors bearing fusion genes between the *AfPAL* and *ArPAL* genes and the enhanced green fluorescent protein gene *eGFP*, *AfPAL/ArPAL-eGFP* (Fig. S2).

## RT-qPCR

A pair of specific primers (5′- AGCAAGATTACGCCTTGCCT-3′/ 5′-AAGGCCTCTACTGCGTTGAC-3′) was designed to amplify a 152 bp fragment of the *AfPAL* and *ArPAL* genes. Another pair of specific primers (5′-CGGGCATTCACGAGACCAC-3′/5′-AATAGACCCTCCAATCCAGACACT-3′) was designed to amplify a 221 bp fragment of the internal reference gene *Actin2* (*Zhang et al., 2012*). The PCR reaction was conducted on SsoFast EvaGreen Supermix (Bio-Rad, Hercules, CA, USA) according to the protocol. The $2^{-\Delta\Delta CT}$ method was used to normalize the expression between the internal reference and the *PAL* genes (*Livak & Schmittgen, 2001*). The data was further analyzed by IBM-SPSS software (http://www-01.ibm.com/software/analytics/spss/).

## Subcellular localization

The 35S-PAL-eGFP recombined vectors were attached onto gold particles ($\varphi = 60\ \mu m$) by the spermidine and $CaCl_2$ method with 35S-eGFP empty vector as control (*Yu et al., 2018*). Onion bulbs were surface sterilized with 75% (v/v) ethanol. The healthy fourth to sixth scales were cut into $2 \times 2$ cm, cultured on Murashige and Skoog's (MS) medium for 4–6 h under dark at 28 °C, and then bombarded using helium biolistic gun (Bio-Rad, USA), incubated for 24 h at 28 °C under dark condition (*Sun et al., 2021*). The fluorescence signal was detected by a confocal microscope (Nikon, Japan; *Yang et al., 2019*).

## Transformation of *Arabidopsis*

The pCAMBIA2300-35S-PAL-eGFP plasmid was mobilized into *Agrobacterium tumefaciens* strain EHA105 and used to transform wild-type *Arabidopsis* as described (*Sun et al., 2020*). The transformed lines were screened on 1/2 MS medium supplemented with 35 mg/L kanamycin (Sigma, St. Louis, MO, USA). The homozygous lines were identified by PCR amplification with specific primers (5′- CATTTGGAGAGGACAGGGTACC-3′/5′-CTAGCAAATAGGGAGAGGAGCTTCA-3′) for Af*PAL* and *ArPAL*.

## Flavonoid quantification

The leaf of $T_3$ lines and wild type were dried and ground to be extracted by 95% alcohol in an ultrasonic instrument at 25 °C for 0.5 h. The extracts were filtered, and the residues were dissolved in 95% alcohol. The residues were filtered again. The filtrates were combined, and solvents were removed under reduced pressure using the rotavapor R-210 (BUCHI, Switzerland) to yield the extract. The template samples were detected using the reagent

color-developing method (NaNO$_2$-Al(NO$_3$)$_3$–NaOH). The above stock solutions 1 ml were added with 0.4 mL 5% NaNO$_2$ for 5 min. The 10% Al(NO$_3$)$_3$ 0.4 ml were added to the reaction. After standing for 5 min, 4% NaOH 4 ml were used to color. The reactions were incubated for 20 min. The quantitative values were determined with a UV-1800 spectrophotometer at 420 nm. The content of total flavonoids was calculated as:

$$\text{Content of flavonoids} = (A_{420} \times V)/(m \times d)$$

where, $A_{420}$ was the absorbance at 420 nm, V represents total volume of the extract, mM was the extraction quality from the leaf of each sample (1 g), d repreents the dilution multiple (*Chen et al., 2007*).

## Anthocyanin measurement

The leaves of T$_3$ lines and wild type were pulverized to fine powder in liquid nitrogen, extracted with acidified (1% HCl) methanol, and incubated dark with shaking for 48 h. Later, it was centrifuged at 4,000 g for 10 min following the protocol described by *Tanaka et al. (1997)*. The supernatant was used to measure absorbance at 535 nm in a UV-1800 spectrophotometer. The anthocyanidin content was indicated by absorption value.

# RESULTS

## Cloning the *AfPAL* and *ArPAL* genes

Based on RNA-seq information, we designed the specific primers to amplify the *AfPAL* and *ArPAL. The* fragments of more than 2,148 bp were amplified from the cDNA library of *A. formosanus* and *A. roxburghii*, respectively (Fig. S3, MK387342 and MK387343). The constructs were verified by PCR and sequencing; the fragment from the cDNA of *A. formosanus* and *A. roxburghii* showed high homology to reported *PAL* bioinformatics-predicted sequences (Fig. S4).

## Proteins sequence analysis

The amino acid sequences of the AfPAL and ArPAL proteins were highly homologous with that from *Phalaenopsis equestris* (XP_020579635.1) and *Dendrobium huoshanense* (Figs. 2 and 3). The AfPAL and ArPAL proteins both contained 715 amino acids with a molecular weight 77.4 kDa, isoelectric point pI 6.18 and 6.26, grand average of hydropathicity (GRAVY) −0.104 and −0.103. The predicted secondary structure of these two proteins contained 48.53% and 48.67% α-helices, 10.07% and 9.79% extended strands, 41.40% and 41.54% random coils, respectively. Their three-dimensional structural model contained all the α-helices, extended strands, and random coils (Fig. S5). Most of these properties of the AfPAL and ArPAL proteins were similar to the PePAL of *Phalaenopsis equestris*.

## Conserved domain and Phylogenetic relationship

The conserved domain of the phenylalanine and histidine ammonia lyase signature (GTITASGDLVPLSYIA) and the active site Ala-Ser-Gly tripeptide forming the MIO group (3,5-dihydro-5-methylene-4H-imidazole-4-one) were found in position 197-213 and 201-203, respectively. Meanwhile, the strictly conserved residues, Y109, L137, S202,
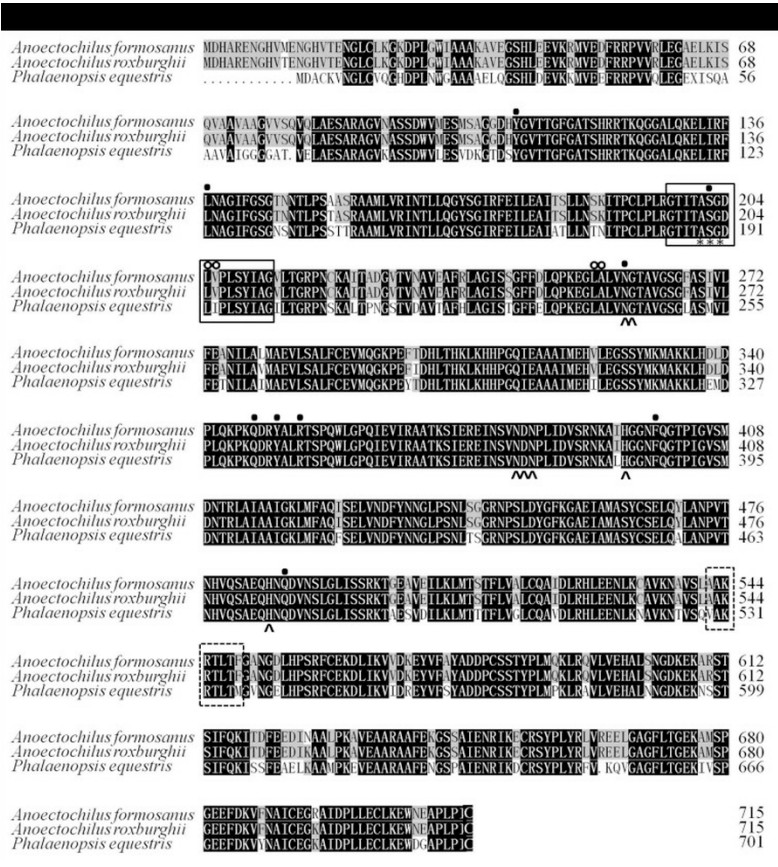

**Figure 2** The structural functional domain of the PAL genes among *A. formosanus*, *A. roxburghii* and *Phalaenopsis equestris*. Identical and conserved amino acid residues are denoted by black (100%), gray (66.6%) and white (0%) backgrounds, respectively. The boxes with solid line represent the phenylalanine and histidine ammonia lyase signature, the boxes with dotted line represent the possible phosphorylation sites, the asterisks represent the active sites, the solid dots represent strictly conserved residues, the circles represent the deamination sites and sharp corners represent the catalytic active sites.

N259, Q347, Y350, R353, F399, and Q487, were found in the AfPAL and ArPAL protein, respectively. Moreover, the deamination sites such as L-205, V-206, L-255, and A-256, catalytic active sites such as N-259, G-260, NDN (381–383 aa), H-395 and HNQDV (485–488 aa), and the possible phosphorylation site such as VAKRVLTF (542–549 aa) were found in both AfPAL and ArPAL (Fig. 2). Multiple alignment and phylogenetic analysis showed that the putative AfPAL and ArPAL proteins were clustered into the same sub-group with the deposited functional PAL proteins of *Dendrobium huoshanense* (Fig. 3), indicating that the PAL proteins from *A. formosanus* and *A. roxburghii* are members of the PAL family.

### Relative expression level under induction

The expression of the *AfPAL* gene in the stem was highest, about twice than the leaf, and it was the 20 times than root from *A. formosanus* ($P \leq 0.05$; Fig. 4). The expression of the *ArPAL* gene in the root was similar to that of the stem, about 10 times higher than

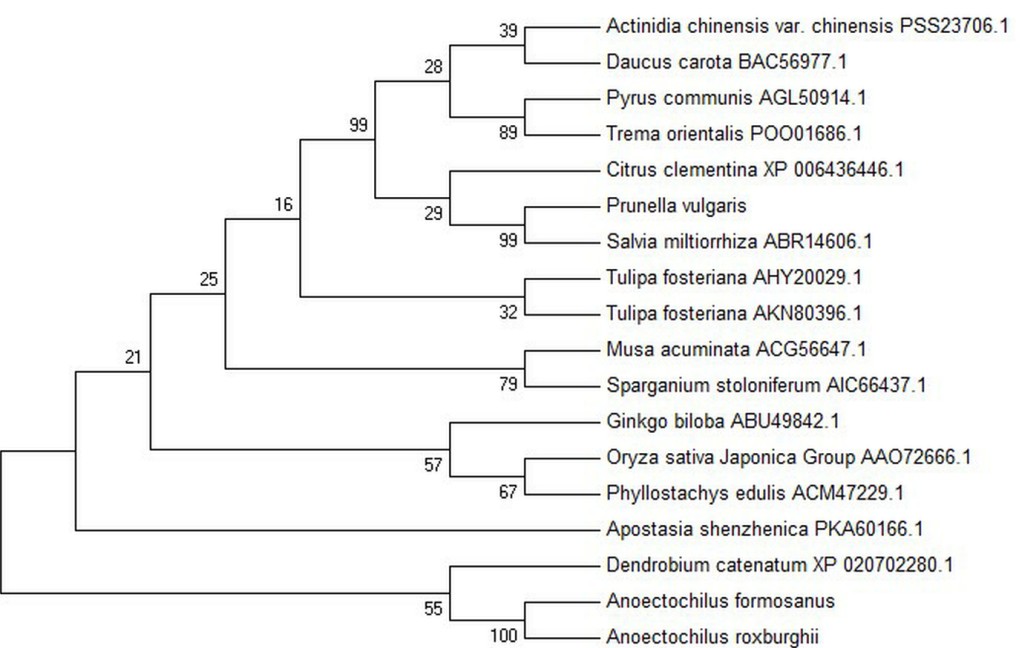

**Figure 3** Phylogenetic tree among the putative proteins of *A. roxburghii*, *A. formosanus* and deposited functional PAL proteins of other plants.

the leaf from *A. roxburghii* ($P \leq 0.05$; Fig. 4). And the expression of the *PAL* gene was similar in root from *A. formosanus*, leaf, and stem from *A. roxburghii*. The expression was downregulated significantly in *A. formosanus* and *A. roxburghii*, and there was a valley value at 1 h in response to Phe stress at the beginning. Then, the expression was upregulated in *A. formosanus* and *A. roxburghii* after 1 h, and reached their peaks at 4 h and 2 h, respectively. Then the expression plummeted, only one-tenth or even less with the control (Fig. 5A). Under the NaCl stress, the expression was upregulated significantly in the two species and reached their valleys at 2 h (15.24 times) and 0.5 h (4.77 times), respectively (Fig. 5B). In response to the UV stress, the overall trend of the expression of the *PAL* genes was upregulated, and there was a peak value at 12 h and 8 h, respectively (54.49 times and 873.89 times, Fig. 5C). The expression was downregulated significantly in *A. formosanus* and reached its valley at 1 h in response to red light stress. In contrast, it was changed but not regular in the *A. roxburghii* (Fig. 5D).

## Subcellular localization
The subcellular localization of the PAL proteins was analyzed using the GFP as a reporter in transient expression assays, and bacterial cells carrying PAL-GFP plasmids were infiltrated into epidermal cells of onion. Confocal microscopy images demonstrated that the PAL-eGFP fusion protein was specifically distributed in the nucleus, whereas GFP alone showed ubiquitous distribution in the whole cell (Fig. 6).

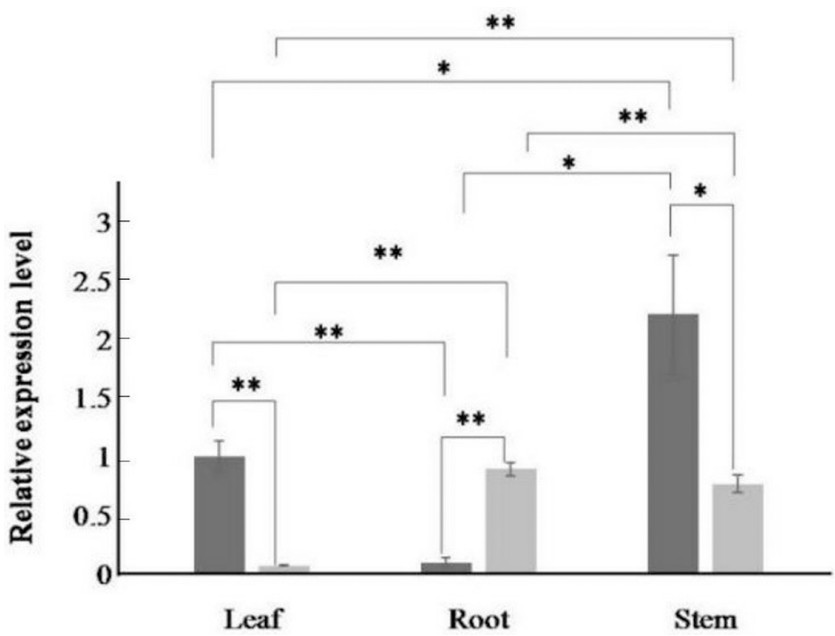

**Figure 4 Relative expression levels of *AfPAL* and *ArPAL* genes among different organs.** The darker columns represent *A. formosanus*, the lighter columns represent *A. roxburghii*. The asterisk (*) and double asterisk (**) stand for significance with the control at 0.05 and 0.01 levels, respectively.

## Overexpression of the *PAL* genes

To investigate the function of *PAL*, the transgenic Arabidopsis of PALs were generated. In the $T_1$ generation, five positive plants of three lines (F1-3) transformed by gene *AfPAL* and two lines (R1-2) by *ArPAL* were screened on the selection medium. In the $T_2$ generation, these lines were a single copy insertion with a ratio of 3:1 between the transformed genes and wild-type *Arabidopsis* (Fig. S6). In the $T_3$ generation, homozygous lines without segregation were identified on the selection medium. The specific PCR amplification confirmed the altered genes from F1-3 and R1-2(Fig. S7). The flavonoid contents were significantly higher in lines F-2 and R-1, and the anthocyanin content was considerably higher in lines F-2, F-3, R-1, and R-2 (Fig. 7). The results revealed that the *PAL* genes were successfully expressed in five independent transgenic events. The flavonoids and anthocyanin contents in transgenic lines were higher than in wild-type.

## DISCUSSION

The ORFs of the cloned *AfPAL* and *ArPAL* genes and the amino acid sequences of their putative proteins were highly homologous with the *PAL* gene and its putative protein in *Dendrobium huoshanense* as well as *Phalaenopsis equestris* (Figs. 2 and 3). The conserved domain of the phenylalanine, histidine ammonia-lyase signature, and the active site Ala-Ser-Gly tripeptide forming the MIO group was necessary for their function (*Gao et al., 2008*). All the active sites were highly conserved among the reported PAL proteins (*Cheng*

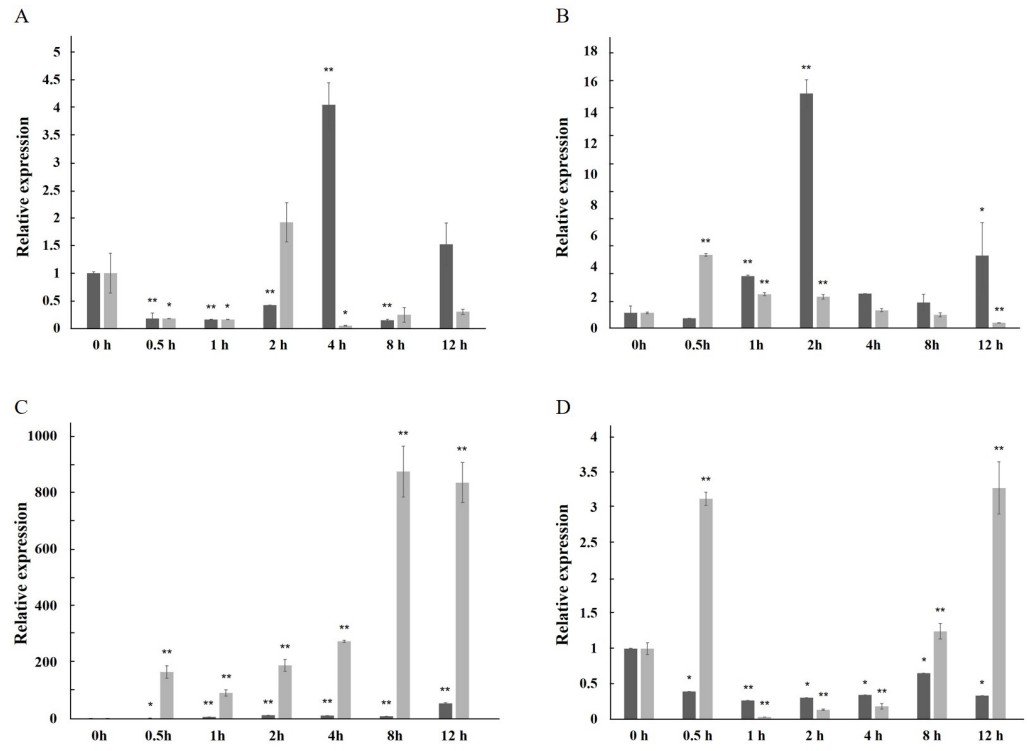

**Figure 5** **Relative expression level of the *PAL* gene under the stress in *A. formosanus* and *A. roxburghii*.** (A) under the Phe stress; (B) under the NaCl stress; (C) under the UV; D: under the red-light stress. The asterisk (*) and double asterisk (**) stand for significance with the control at 0.05 and 0.01 levels, respectively.

*et al., 2005*; *Gao et al., 2008*; *Jin et al., 2013*; *Li et al., 2012*; *Zhang et al., 2016a*; *Zhang et al., 2016b*).

In many other plants, the expression of the *PAL* genes showed organic specificity, and the expression level was correlated to their accumulation of flavonoids (*Fukasawa-Akada, Kung & Watson, 1996*; *Jin et al., 2013*; *Leyva et al., 1992*; *Zhang et al., 2016a*; *Zhang et al., 2016b*). In this paper, RT-qPCR analysis revealed that the high expression of the *PAL* genes was found in the stem of *A. formosanus* and the root of *A. roxburghii* (Fig. 4). However, their differential expression was responsive to four stress or induction treatments. This result implies the different tolerance of these two species in the activities of *PAL* (Fig. 5). The expression of the *PAL* genes was intensely upregulated in response to NaCl and UV (Figs. 5B and 5C), which was consistent with observations of other plants under stress conditions (*Bell et al., 2017*; *El, Wilson & Callahan, 2003*; *El-Shora, 2002*; *Fritzemeier & Kindl, 1981*; *Song & Wang, 2009*). The range of the differential expression of the *AfPAL* gene was more extensive than the *ArPAL* gene under NaCl stress, but the content of the *AfPAL* gene was less than the *ArPAL* gene, conversely. In response to Phe and red-light induction, the expression of the *PAL* gene was downregulated, excepting a few sharp peaks (Figs. 5A and 5D). Similar results were found in other plants (*Nakazawa et al., 2001*; *Bellini & Hillman, 1971*; *Edahiro et al., 2005*; *Heo et al., 2012*). Therefore, it is preliminary concluded that the

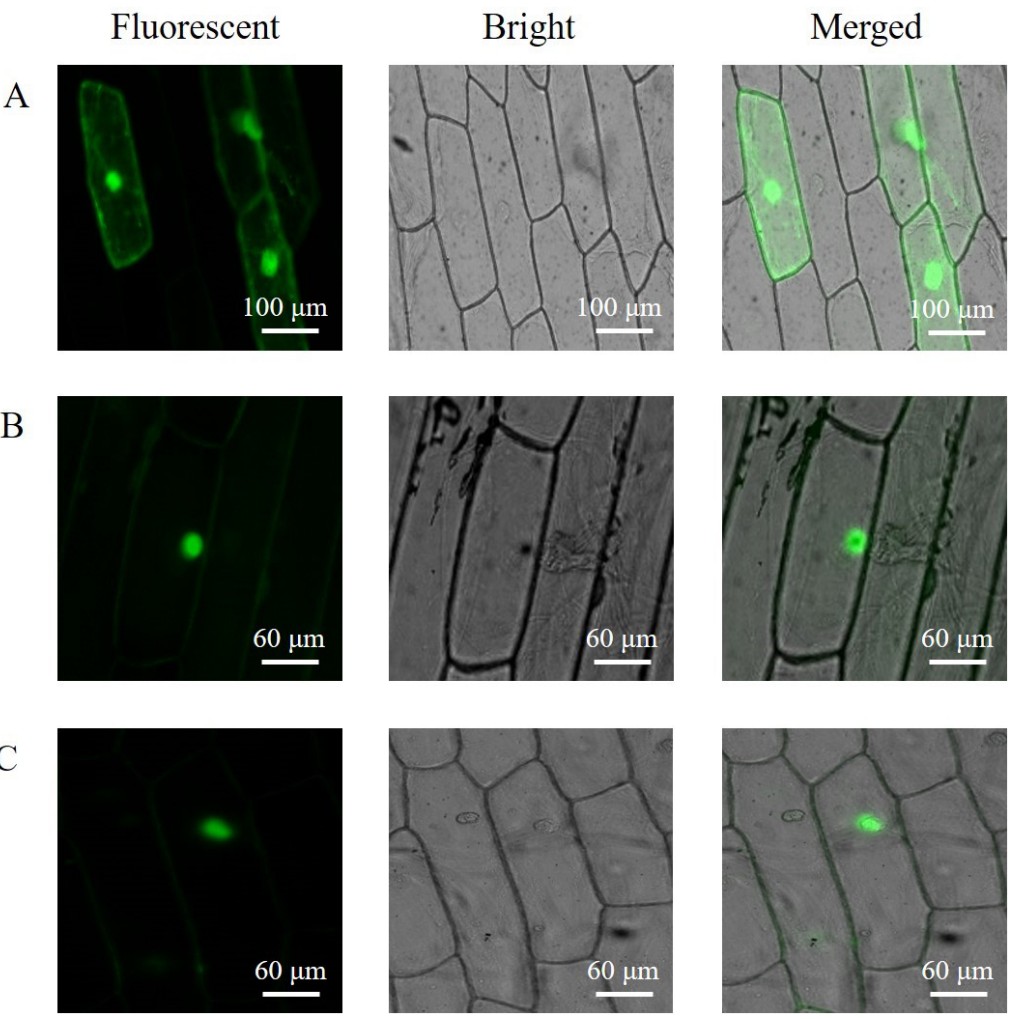

**Figure 6  Subcellular localization of PAL protein.** *eGFP* and *PAL-eGFP* fusion gene were driven under the control of the CaMV 35 Spromoter. (A) Epidermal cells of onion transformed by pC2300-*35S-eGFP*. (B) Epidermal cells of onion transformed by pC2300-*35S-PAL-eGFP* from *A. formosanus*. (C) Epidermal cells of onion transformed by pC2300-35S-PAL-eGFP from *A. roxburghii*.

expression of the *PAL* gene is more sensitive to saline induction in *A. formosanus* than *A. roxburghii*, and the latter is probably more sensitive to UV induction.

Subcellular localization of PAL protein has been studied in different plants (*Fukasawa-Akada, Kung & Watson, 1996*; *Herdt & Wiermann, 1978*). The present study investigated the subcellular localization of PAL protein in a heterologous system (the chloroplast-free epidermal cells of onion) by confocal laser-scanning microscopical imaging of GFP fluorescence (Fig. 6). Transient expression of the *PAL-eGFP* fusion protein in the onion was targeted to the nucleus. The nuclei of five tree species with respect to the presence of flavanols (*Feucht, Treutter & Polster, 2014*). Flavonoids and at least two of the biosynthetic

A

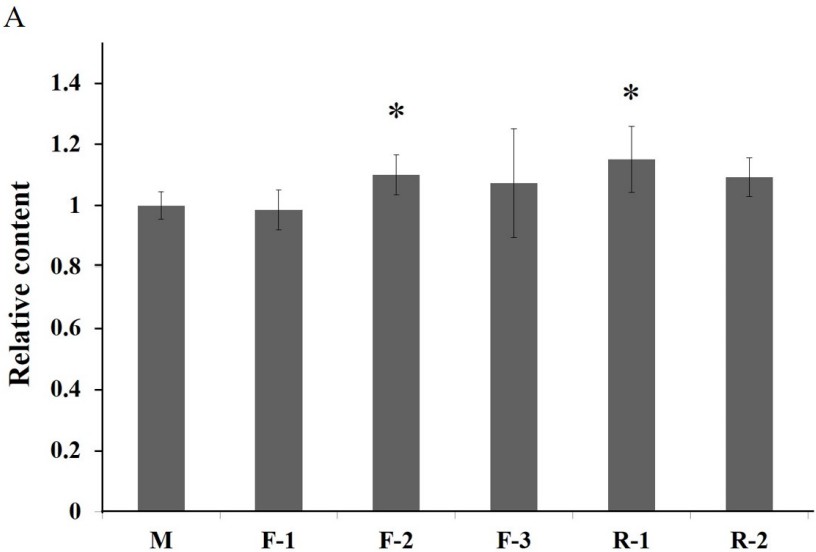

B

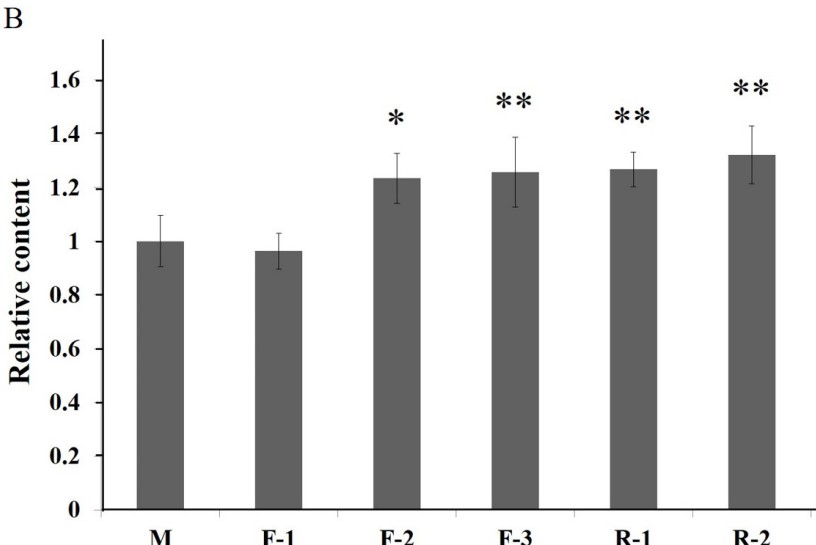

**Figure 7** **Relative content of total flavonoids and anthocyanin of T3 *Arabidopsis* lines of gene *PAL* from *A. formosanus* and *A. roxburghii*.** (A) Relative content of total flavonoids of T3 *Arabidopsis* lines of gene *PAL* from *A. formosanus* and *A. roxburghii*. (B) Relative content of anthocyanin of T3 *Arabidopsis* lines of gene *PAL* from *A. formosanus* and *A. roxburghii*. The asterisk (\*) and double asterisk (\*\*) stand for significance with the control at 0.05 and 0.01 levels, respectively.

enzymes are located in the nucleus in several cell types in *Arabidopsis* (*Saslowsky & Winkel-Shirley, 2005*). The result might indicate a high association of PAL protein to the nucleus or nuclear membrane and raise the possibility of novel mechanisms of action for flavonoids.

In the overexpressing transgenic lines of *Arabidopsis thaliana*, the content of flavonoids and anthocyanin was significantly higher than those in wild-type control (Fig. 7). The increased contents of total flavonoids should be associated with the genetically modified

anthocyanin metabolic pathway (Fig. S1). In addition, the content of total flavonoids and anthocyanins was higher in the *Arabidopsis* lines transformed by the *ArPAL* gene than those of the *AfPAL*. This result suggests that the activities of the proteins encoded by *PAL* genes might be differential between these two species. The transgenic tobacco with the overexpression *PAL* gene was developed in response to infection by tobacco mosaic virus and necrotrophic pathogens (*Pallas et al., 2010*; *Way, Birch & Manners, 2011*). Many reports indicated its critical function in the secondary metabolism of these plants (*Lois et al., 1989*; *Nugroho, Verberne & Verpoorte, 2002*).

## CONCLUSIONS

The *AfPAL* and *ArPAL* genes' expression showed organic specificity and the differential expression of the *PAL* genes in response to four treatments. The flavonoid metabolites of *Arabidopsis* transformed with *AfPAL*, and the *ArPAL* gene were increased, provided by the anthocyanin metabolic pathway. And the different effects of overexpressed *Arabidopsis* flavonoids were caused by different *Anoectochilus* of the *PAL* gene.

### Funding

This research was funded by the Natural Science Foundation of Fujian Province, China (2020J01382) the Education Scientific Fund for Young Teachers from Fujian Education Department (JAT190703) and the Sanming Science and Technology Leading Project (2019-S-32 and 2020-S-7). The funders had no role in study design, data collection and analysis, decision to publish, or preparation of the manuscript.

### Grant Disclosures

The following grant information was disclosed by the authors:
Natural Science Foundation of Fujian Province, China: 2020J01382.
Education Scientific Fund for Young Teachers from Fujian Education Department: JAT190703.
Sanming Science and Technology Leading Project: 2019-S-32, 2020-S-7.

### Competing Interests

The authors declare there are no competing interests.

### Author Contributions

- Lin Yang conceived and designed the experiments, performed the experiments, prepared figures and/or tables, authored or reviewed drafts of the article, and approved the final draft.
- Wan-Chen Li analyzed the data, authored or reviewed drafts of the article, and approved the final draft.
- Feng-ling Fu analyzed the data, authored or reviewed drafts of the article, and approved the final draft.

- Jingtao Qu performed the experiments, prepared figures and/or tables, and approved the final draft.
- Fuai Sun performed the experiments, prepared figures and/or tables, and approved the final draft.
- Haoqiang Yu performed the experiments, authored or reviewed drafts of the article, and approved the final draft.
- Juncheng Zhang conceived and designed the experiments, prepared figures and/or tables, and approved the final draft.

## DNA Deposition

The following information was supplied regarding the deposition of DNA sequences:

The sequences of PAL from *Anoectochilus formosanus* and *Anoectochilus roxburghii* are available at GenBank: MK387342 and MK387343.

## Data Availability

The raw measurements are available in the Supplementary File.

## Supplemental Information

Supplemental information for this article can be found online at http://dx.doi.org/10.7717/peerj.13614#supplemental-information.

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
