# Peer review of "Characterization of phenylalanine ammonia-lyase genes facilitating flavonoid biosynthesis from two species of medicinal plant Anoectochilus"

_PeerJ, doi:10.7717/peerj.13614_

## Round 0.1 · original submission · Major Revisions

Please provide a comprehensively revised version addressing the editorial comments and a detailed rebuttal letter.

Reviewer 1 ·

Basic reporting

no comment.

Experimental design

Materials and methods need to be improved.
1. The treatment of (L-Phe), NaCl, UV and red-light is not specified in the method.
2. Is the determination of flavonoids supported by references?

Validity of the findings

1. Why is there no amino acid sequence of Dendrobium huoshanense in the sequence alignment of Figure 3?
2.In the overexpression experiment, the total flavonoid content of F-1, F-3 and R-2 did not reach a significant level. Is this result reliable? In Figure 8, the effect of PAL on the synthesis of anthocyanin is more significant, and the expression of anthocyanin biosynthesis genes needs to be determined.

Additional comments

1. The first letter of the on line 150 should be capitalized.
2. Punctuation after line 152 (1997).
3. The length of the PAL fragment in Figure 2 is 2148bp, and the marker is preferably 5000.
4. In Figure 5, Figure 6 and Figure 8, it is best to directly mark what the columns of different colors represent in the pictures, and please pay attention to the position of the '*' representing significance in Figure 6.
5.The discussion needs to be further improved. There should be a space between the and differentail on line 237.

Reviewer 2 ·

Basic reporting

no comment

Experimental design

no comment

Validity of the findings

no comment

Additional comments

1) The level of English throughout your manuscript does not meet the journal’s required standard. There are many sentence errors in the paper, which makes the reader unable to read smoothly.
2) The function of PAL in Anoectochilus has not been deeply studied, and the paper lacks depth and innovation.
3) The ruler is missing in the Figure 1,7.
4) Clear legends are missing in figures 6 and 8.
5) Figure 7 -C the image is not clear and the image is deformed.

Reviewer 3 ·

Basic reporting

The paper Characterization of phenylalanine ammonia-lyase genes facilitating flavonoid biosynthesis from two species of medicinal plant Anoectochilus presents the cloning and expression of PAL genes from two related species used in traditional medicine. Although the paper is very descriptive some relevant results are presented. Authors do present assays about localization of gene products presented in this manuscript, therefore with major changes the paper can be more engaging The readability of the discussion could be improved.
The English needs to be improved through the manuscript.
Figure legends can be improved too, some suggestions are given at the manuscript for example figure 2 can be moved to supplementary figures.
Some suggested changes are the following:
Line 18, and have been used
Line 22, PAL genes were cloned
Line 24, used for subcellular
Line 26, stresses and were analyzed
Line 28, base pairs and were
Line 29, that both
Lines 32-33, could you please make this sentence more clear?
Line 43, obviously different
Line 48, is it imperative to promote accumulation?
Line 50, primary and the
Line 68, stresses
Line 76, surface
Line 128, separate microscope model from references
Line 133, were identified
Line 137, to be extracted
Line 138, extracts were filtered and the residues were dissolved
Lines 141-142, sentence is not clear
Line 168, theoretical secondary structure?
Line 187, about twice it is two times, however it says 20 times, could you please explain
Others suggestions are given at the pdf file of the reviewed manuscript.

Experimental design

Although the research is within the journal scope, the materials and methods section could be improved, by adding information missing (stated at the reviewed version in comments). Statistical analysis is not presented at this section. It is important to describe the analysis that were used in the expression analysis, comparisons among samples.

Validity of the findings

The manuscript is not engaging in the way that is described, since at introduction, materials and methods some sentences are not presented in a clear way. At the discussion section the contribution of knowledge should be stated. Conclusions need to be more clear.

Additional comments

The manuscript can be improved making sentences more clear and complete. The alignment figure is not of good quality.

Annotated reviews are not available for download in order to protect the identity of reviewers who chose to remain anonymous.

---

## Round 0.2 · Minor Revisions

Please be encouraged by the reviews, and we look forward to your revised manuscript.

Reviewer 1 ·

Basic reporting

No comment.

Experimental design

No comment.

Validity of the findings

No comment.

Additional comments

There are many formats that need to be modified in the manuscript.
1. 93 lines of punctuation.
2. Up-regulated in line 257 should be changed to upregulated.
3. The gene in line 284 should be italicized.
4. The gene in line 290 should be oblique. apostrophe?
5. References in lines 300-301, 317-318, 338-339, 348-349, 356-357, and 377-378 should be lowercase except for the first letter of the first word.
6. Nicotiana on line 327 should be italicized.
7. Protoplasma on line 337 should be italicized.
8. Plant Physiology on line 342 should be italicized.
9. The EMBO Journal on line 355 should be italicized.
10. Plant Physiology and Biochemistry in line 361 should be italicized.
11. The Journal of Biomedical Science on line 372 should be italicized.
12. Genes & Genetic Systems on line 384 should be italicized.
13. Plant Growth Regulation on line 395 should be italicized.
14. The Journal of Separation Science on line 399 should be italicized.
15. PAL on line 407 should be italicized.

Reviewer 3 ·

Basic reporting

The manuscript is now clear and language professional. The authors made the suggested corrections at the different parts like introduction, results and discussion.

Experimental design

The authors improved this part and added statistical analysis. The description of materials and methods is much better and information is enough to replicate.

Validity of the findings

The findings presented in a clearer way are robust. Changes were made and figures look much better. Novel work is presented in the two species of traditional medicine. The importance of PAL genes in flavonoid biosynthesis is strengthened.

Additional comments

The manuscript is now more engaging and readable.

---

## Round 0.3 · Minor Revisions

There are still problems with the language quality - starting with the first sentence of the abstract.

Please have the manuscript corrected and edited by a professional editor and provide documentation of this when you resubmit.

---

## Round 0.4 · accepted · Accept

Thanks, your manuscript is accepted at PeerJ.